# Laquinimod Supports Remyelination in Non-Supportive Environments

**DOI:** 10.3390/cells8111363

**Published:** 2019-10-31

**Authors:** Stella Nyamoya, Julia Steinle, Uta Chrzanowski, Joel Kaye, Christoph Schmitz, Cordian Beyer, Markus Kipp

**Affiliations:** 1Institute of Anatomy, Rostock University Medical Center, 18057 Rostock, Germany; Stella.Nyamoya@rwth-aachen.de; 2Institute of Neuroanatomy and JARA-BRAIN, Faculty of Medicine, RWTH Aachen University, 52074 Aachen, Germany; Julia.Steinle@rwth-aachen.de (J.S.); CBeyer@ukaachen.de (C.B.); 3Department of Anatomy II, Ludwig-Maximilians-University of Munich, 80336 Munich, Germany; Uta.Chrzanowski@med.uni-muenchen.de (U.C.); Christoph_Schmitz@med.uni-muenchen.de (C.S.); 4AyalaPharma, VP Research & Nonclinical Development, Rehovot 7670104, Israel; joel.k@ayalapharma.com; 5Centre for Transdisciplinary Neurosciences, Rostock University Medical Center, 18057 Rostock, Germany

**Keywords:** multiple sclerosis, remyelination, cuprizone, neurodegeneration, laquinimod

## Abstract

Inflammatory demyelination, which is a characteristic of multiple sclerosis lesions, leads to acute functional deficits and, in the long term, to progressive axonal degeneration. While remyelination is believed to protect axons, the endogenous-regenerative processes are often incomplete or even completely fail in many multiple sclerosis patients. Although it is currently unknown why remyelination fails, recurrent demyelination of previously demyelinated white matter areas is one contributing factor. In this study, we investigated whether laquinimod, which has demonstrated protective effects in active multiple sclerosis patients, protects against recurrent demyelination. To address this, male mice were intoxicated with cuprizone for up to eight weeks and treated with either a vehicle solution or laquinimod at the beginning of week 5, where remyelination was ongoing. The brains were harvested and analyzed by immunohistochemistry. At the time-point of laquinimod treatment initiation, oligodendrocyte progenitor cells proliferated and maturated despite ongoing demyelination activity. In the following weeks, myelination recovered in the laquinimod- but not vehicle-treated mice, despite continued cuprizone intoxication. Myelin recovery was paralleled by less severe microgliosis and acute axonal injury. In this study, we were able to demonstrate that laquinimod, which has previously been shown to protect against cuprizone-induced oligodendrocyte degeneration, exerts protective effects during oligodendrocyte progenitor differentiation as well. By this mechanism, laquinimod allows remyelination in non-supportive environments. These results should encourage further clinical studies in progressive multiple sclerosis patients.

## 1. Introduction

Multiple sclerosis (MS) has a complex pathomechanism, leading to the formation of inflammatory demyelinating lesions within the central nervous system (CNS). Such lesions, which can be found in white and grey matter, are characterized by the loss of oligodendrocytes to a variable extent, focal and diffuse myelin pathology, and astrocyte and microglia activation, as well as damage to nerve cells [1,2,3,4,5,6]. On the clinical level, distinct disease courses can be distinguished: At the beginning of the disease most patients suffer from the sudden occurrence of new neurological symptoms, which usually disappear after several weeks [7]. This initial disease course is called relapsing-remitting MS (RRMS), which means that symptoms appear (i.e., a relapse) and then fade away, either partially or completely (i.e., remitting). By definition, during the RRMS disease phase, the level of clinical disability remains stable in between two relapses. After several years (10–15 years), the frequency of relapses decreases, and patients clinically deteriorate independent of the relapses. This so-called secondary-progressive MS (SPMS) course is characterized by chronically progressive clinical worsening over time, with or without superimposed relapses. In about 15% of patients, the disease is characterized by neurologic worsening (accumulation of disability) from the onset of symptoms, without early relapses or remissions. Here, this is called primary-progressive MS (PPMS). Depending on the location of the demyelinated lesions within the CNS, clinical symptoms can vary substantially between different patients [8,9]. While inflammation is believed to be the pathological correlate of relapses during RRMS, neurodegeneration, especially axonal damage, is thought to be the underlying substratum of irreversible clinical disability accumulating during the progressive stage of the disease [10,11].

The mechanisms underlying the progressive neurodegeneration in MS are currently unknown, but the failure of remyelination appears to play a significant role. Remyelination is a very complex biological process and can be classified, on the cellular level, as four consecutive steps: (i) Proliferation of oligodendrocyte progenitor cells (OPCs); (ii) OPC migration towards the demyelinated axons; (iii) OPC differentiation; and finally, (iv) interaction of the premature oligodendrocyte with the axon (i.e., axon wrapping) [12]. It has been accepted since early neuropathological studies that in MS, demyelinated lesions can remyelinate, although there are extremely limited data on the quantitative extent and natural history of this repair process [13,14,15]. The existence of so called ‘shadow plaques’, representing remyelinated lesions, clearly demonstrates that the complete repair of MS plaques is principally possible, although it is more common to observe only limited repair at the edge of lesions [16,17]. It is not clear why in some patients remyelination is widespread while in others it is sparse. Clearly, remyelination is neuroprotective, and if remyelination fails, different biochemical mechanisms can trigger delayed axonal degeneration, along with an increased energy demand of impulse conduction along excitable demyelinated axons [18], a lack of axonal trophic support provided by oligodendrocytes [19], a lethal rise in intra-axonal calcium levels [20], or the higher vulnerability of demyelinated axons against cytotoxic substances. Of note, several reports suggest that the recurrent demyelination of demyelinated white matter areas is one of the factors underlying remyelination failure in MS [21,22,23]. For example, in a detailed histopathological study, around 15% of remyelinated shadow plaques showed evidence of a superimposed, new demyelinating activity [21]. In another study, serial magnetization transfer imaging in RRMS and SPMS patients showed that in both patient cohorts new lesions form in previously lesional tissue that appears to be experiencing a second round of inflammatory demyelination (i.e., repeat lesions) [23]. In line with these findings, several pre-clinical studies have shown that recurrent episodes of demyelination occur at sites of previous demyelination [24,25,26]. Indeed, demyelinated foci can be a potent trigger for peripheral immune cell recruitment [27,28,29]. An increased understanding of how inflammation alters cell intrinsic and extrinsic factors to, in turn, influence OPC proliferation, recruitment, differentiation, and, ultimately, remyelination, is crucial for the development of novel therapies that target disease progression. Moreover, a need exists to elucidate the factors that contribute to successful remyelination as well as those that result from its failure.

One of the models which allows such questions to be addressed is the cuprizone model. In this model, oral intoxication with copper-chelator cuprizone induces, oligodendrocyte apoptosis within days, which is closely followed by the activation of the innate immune cells of the brain, i.e., astrocytes and microglia, finally leading to the demyelination of distinct white and grey matter brain areas. Although OPCs are activated during the course of cuprizone intoxication [30], remyelination fails due to an ongoing cuprizone-induced mature oligodendrocyte injury. In this study, we tested the hypothesis that laquinimod, which has been shown to ameliorate cuprizone-induced demyelination [31,32], supports remyelination in non-supportive environments. To this end, mice were intoxicated with cuprizone for up to 8 weeks and treated with laquinimod, starting at the beginning of week 5, when the endogenous remyelination pathways are activated in this model. We can show that daily laquinimod treatment induces myelin recovery, which is paralleled by an amelioration of axonal degeneration.

## 2. Material and Methods

### 2.1. Animals and Experimental Setup

C57BL/6 mice were purchased either from Janvier Labs, Le Genest-Saint-Isle, France or provided by TEVA. All experimental procedures were approved by the Review Board for the Care of Animal Subjects of the district government (Regierung Oberbayern; reference number 55.2-154-2532-73-15; Germany) and the local ethics committee of Teva Pharmaceutical Industries Ltd. The animals were randomly allocated to the different experimental groups and were kept under standard laboratory conditions with access to food and water ad libitum. Demyelination was induced by intoxicating mice with a diet containing 0.3% cuprizone (bis(cyclohexanone)oxaldihydrazone) mixed into pellets and or standard pelleted rodent chow for the indicated treatment period. The control mice (5 animals per group) were fed standard pelleted rodent chow for the entire duration of the study. The following treatment groups were included in the study: Control animals (CO), 4-weeks CPZ (cuprizone), where the animals were intoxicated with cuprizone for 4 weeks; 6-week CPZ and vehicle, where the animals were intoxicated with cuprizone for 6 weeks. With this group, at the beginning of week 5, the animals were subjected daily to an oral gavage with the vehicle solution (100 µL); 6-week CPZ and laquinimod (LAQ), where the animals were intoxicated with cuprizone for 6 weeks. With the group, at the beginning of week 5, the animals were subjected daily to an oral gavage with the laquinimod solution (100 µL); 8-week CPZ and vehicle, where the animals were intoxicated with cuprizone for 8 weeks. With this group, at the beginning of week 5, the animals were subjected daily to an oral gavage with the vehicle solution (100 µL); 8-week CPZ and LAQ, where the animals were intoxicated with cuprizone for 8 weeks. With this group, at the beginning of week 5, the animals were subjected daily to an oral gavage with the laquinimod solution (100 µL). See Figure 1A for a schematic illustration of the experiment setup. An additional cohort of C57BL/6 mice was intoxicated with 0.25% cuprizone mixed in the ground rodent chow for the duration of 1 (5 animals), 3 (4 animals), or 5 weeks (5 animals) to study OPC responses.

### 2.2. Tissue Preparation

For the histological and immunohistochemical studies, the preparation of tissue was performed as previously described [33,34]. In brief, mice were transcardially perfused with ice-cold PBS (Phosphate-buffered saline), followed by a 3.7% formalin solution (pH 7.4). After overnight post-fixation in the same fixative, the brains were dissected and embedded in paraffin, and coronal 5-μm-thick sections were prepared for immunohistochemistry. The coronal slices were analyzed at level 265 according to the mouse brain atlas published by Sidman et al. (http://www.hms.harvard.edu/research/brain/atlas.html).

### 2.3. Luxol Fast Blue (LFB) Periodic Acid–Schiff (PAS) Stain and Myelin Status Scoring

The intact and damaged myelin were both histochemically visualized using Luxol fast blue / periodic acid–Schiff (LFB/PAS) stains. To this end, the slides were deparaffinized in 4 × 5 min xylene, rinsed 3 × 3 min in 100% ethanol, followed by 2 × 5 min in 96% ethanol. The sections were then subsequently incubated in a LFB solution (0.1 g Luxol fast blue; 7709, Carl Roth, Karlsruhe, Germany) in 100 mL 96% ethanol plus 500 μL acetic acid (3738, Carl Roth, Germany), overnight at 60 °C. On the next day, the sections were dipped into 96% ethanol, followed by water, and processed in a lithium carbonate solution (0.05 g lithium carbonate [1.05680.0250, Merck, Darmstadt, Germany] in 100 mL aqua (dist.). The sections were further differentiated in 70% ethanol for a few seconds and rinsed in water. Afterwards, oxidation was performed in periodic acid (0.5 g periodic acid [1.00524.0025, Merck, Germany] in 100 mL aqua dist.). Sections were rinsed, followed by incubation in Schiff’s reaction (1.09033.0500, Merck, Germany) for 15 min, then rinsed in warm tap water for 5 min and counterstained with hematoxylin (1.04302.0025, Merck, Germany) for 1 min. The sections were dehydrated and subsequently mounted in DePeX (18243, Serva Electrophoresis GmbH, Heidelberg, Germany) for further analyses. Myelination in the corpus callosum was analyzed by scoring the LFB/PAS stained sections, ranging from 100% (“normal” myelination) to 0% (complete demyelination). Two evaluators blinded to the treatment groups performed the scoring, and the results were averaged.

### 2.4. Immunohistochemistry and Densitometric Analyses

For immunohistochemistry, sections were rehydrated and, if necessary, antigens were unmasked with heat in a Tris/EDTA (pH 9.0) or citrate (pH 6.0) buffer. After washing in PBS, sections were incubated overnight (4 °C) with the different primary antibody solutions (Table 1). The following primary antibody concentrations were applied: Anti-PLP 1:5000, anti-MAG 1:2500, anti-IBA1 1:10000, anti-APP 1:5000, anti-APC/CC1 1:250, and anti-OLIG2 1:2000. The next day, the slides were incubated in a biotinylated secondary antibody solution [(i) horse anti-mouse IgG, 1:50; (ii) goat anti-rabbit IgG, 1:50] for 1 h and then incubated in a peroxidase-coupled avidin–biotin complex solution (ABC-HRP kit; PK-6100, RRID AB 2336819, Vector Laboratories, USA). Finally, the slides were incubated in 3,3′-diaminobenzidine (K3468, DAKO, Germany) as a peroxidase substrate. A detailed list of applied antibodies is given in Table 1 and Table 2.

The stained and processed sections were digitalized using a Nikon Eclipse 80i microscope (Nikon Instruments, Düsseldorf, Germany) equipped with a DS-2MV camera. The open source program ImageJ 1.48v (NIH, Bethesda, MD, USA) was used to evaluate the staining intensities using semi-automated densitometrical evaluation after a threshold setting. Relative staining intensities were then semi-quantified in binary converted images, and the results were presented as percentage areas. Cell and spheroid numbers were quantified after manually delineating the region of interest (ROI; i.e., the midline of the corpus callosum). The ROI-area was quantified, and results were presented as cells or spheroids/mm^2^. Additional slides were scanned using the Nikon Eclipse E200 microscope (Nikon Instruments, Germany) equipped with a Basler acA1920-40um camera (Basler AG, Ahrensburg, Germany) and manual scanning software (manualWSI software, Microvisioneer, München, Germany). Cell numbers were quantified after manually delineating the ROI using the program ViewPoint (PreciPoint GmbH, Freising, Germany). Two evaluators blinded to the treatment groups performed the scoring and the results were averaged.

### 2.5. Immunofluorescence Double Labelling

For the immunofluorescence double labeling experiments, the sections were rehydrated, the sites of antigens unmasked by heating in a Tris/EDTA (pH 9.0) or citrate (pH 6.0) buffer, blocked with PBS containing 2% heat-inactivated fetal calf serum ([FCS], A15-152, PAA, Germany) and 1% bovine serum albumin ([BSA], 0163, Carl Roth, Karlsruhe, Germany), and incubated overnight (4 °C) with the first primary mouse anti-OLIG2 antibody (1:1000) diluted in the blocking solution to visualize oligodendrocytes. After washing, the sections were incubated with the appropriate fluorescent secondary antibody (1:500, anti-mouse Alexa Fluor 546) diluted in the blocking solution for 1 h (room temperature). The sections were subsequently washed and incubated overnight (4 °C), with the second rabbit anti-Ki67 primary antibody (1:1500) diluted in the blocking solution to visualize proliferating cells. After washing, the sections were incubated with the second fluorescent secondary antibody (1:500, anti-rabbit Alexa Fluor 488) diluted in the blocking solution for 1 h. Subsequently, sections were incubated with a Hoechst 33342 solution (1:10000, H3570, Thermo Fisher Scientific, Waltham, MA, USA) diluted in PBS for the staining of cell nuclei. A detailed list of applied antibodies is given in Table 1 and Table 2. The stained and processed sections were documented with the Leica microscope DMI6000B working station (Leica Microsystems, Wetzlar, Germany). Cell numbers were quantified after manually delineating the corpus callosum using the open source program ImageJ 1.48v (NIH, USA). Single and double positive cells were counted within the region of interest. Two evaluators (S.N. and J.S./U.C.) blinded to the treatment groups performed the scoring, and the results were averaged. To rule out unspecific binding of the fluorescent secondary antibodies to primary antibodies, appropriate negative controls were performed by first incubating sections with the primary antibodies and subsequently incubating these sections with the wrong fluorescent secondary antibody. Unspecific secondary antibody binding to the tissue itself was checked by performing negative controls by incubating sections with each of the fluorescent secondary antibodies alone (data not shown).

### 2.6. Statistical Analyses

The statistical analyses were performed using GraphPad Prism 5. The data are presented as arithmetic means ± SEM. Non-Gaussian distribution was assumed. The data were analyzed with either the Mann–Whitney or Kruskal–Wallis test, followed by Dunn’s multiple comparison test. Here, results where *p* < 0.05 were considered statistically significant. The following symbols were used to indicate the level of significance: * *p* < 0.05, ** *p* < 0.005, and *** *p* < 0.001.

## 3. Results

To determine the potential effect of laquinimod (LAQ) on intrinsic remyelination capacity during continuous toxin-induced demyelination, mice were intoxicated with cuprizone (CPZ) for either 6 or 8 weeks. At the beginning of week 5, both cohorts were treated with either the vehicle or laquinimod solution until the end of the experiment (see Figure 1A for the detailed experimental setup).

To analyze the extent of de- and re-myelination, LFB/PAS, anti-PLP, and anti-MAG stains were performed, and the staining intensity was quantified in the midline of the corpus callosum (Figure 1D,E). At the end of week 4, myelin pathology was clearly evident in the LFB/PAS stained sections, indicated by a pronounced loss of LFB staining intensity (Figure 1B). Anti-PLP (co 95.1 ± 1.05%, 4 wks CPZ 69.9 ± 4.26%) and anti-MAG (co 68.9 ± 8.56%, 4 wks CPZ 30.0 ± 4.52%) staining intensity loss was less severe, yet clearly evident. To analyze, whether laquinimod protects new-born myelinating oligodendrocytes, and thus support endogenous remyelination in a non-supportive environment, another cohort of mice was treated daily with either the vehicle or laquinimod solution, starting at the beginning of week 5. To allow early remyelination, mice were sacrificed at week 6. As demonstrated in Figure 1, the anti-PLP and anti-MAG staining intensities were higher in the laquinimod-treated mice compared to the vehicle-treated mice. To verify these findings, a third cohort of mice was sacrificed at week 8 (i.e., after 4 weeks of laquinimod treatment), and the myelination status of the corpus callosum was analyzed. Comparable to what we found at week 6, the anti-PLP and anti-MAG staining intensities were higher in the laquinimod-treated mice compared to the vehicle-treated mice.

To analyze, whether higher myelination levels in laquinimod-treated groups are paralleled by higher densities of mature oligodendrocytes, we quantified the densities of anti-APC^+^ cells in the different treatment groups (Figure 1F). A severe reduction of APC^+^ cell densities was found in both vehicle treated groups (co 932.8 ± 8.04 cells/mm^2^, 6 weeks CPZ and Veh 261.5 ± 15.34 cells/mm^2^, 8 weeks CPZ and Veh 188.6 ± 31.41 cells/mm^2^), whereas the number of APC^+^ cells were significantly higher in the laquinimod-treated groups (6 weeks CPZ and LAQ 420.7 ± 21.78 cells/mm^2^, 8 weeks CPZ and LAQ 240.9 ± 24.32 cells/mm^2^).

To verify whether remyelination was ongoing at the time point when we started the laquinimod treatment (i.e., at the beginning of week 5), we analyzed the densities of proliferating OPC in OLIG2/Ki67-double stained sections during the course of cuprizone-induced demyelination. As demonstrated in Figure 2A, the number of OLIG2^+^/Ki67^+^ cells, resembling proliferating OPCs, were low in control animals and animals intoxicated for 1 week with cuprizone, but high (~70 cells/mm^2^) at weeks 3 and 5. The percentage of Ki67^+^ cells among all OLIG2^+^ cells was highest at week 3 and decreased till week 5 (Figure 2B). Furthermore, the number of APC^+^ cells was low at weeks 1 and 3 but recovered at week 5 (Figure 2D). These results implicate that remyelination was already ongoing at week 3 and entered the OPC differentiation stage at week 5. Taken together, these results suggest that while remyelination fails during a continuous cuprizone-intoxication protocol in vehicle-treated mice, myelin recovery occurs in laquinimod-treated mice.

The recent work of our lab suggests that the extent of microglia activation is an important determinant for acute axonal injury in this model [35]. To analyze whether the activation status of microglia cells was affected by the laquinimod treatment, anti-IBA1 stains were performed, and differences in staining intensities were assessed via densitometric analyses. As demonstrated in Figure 3A,C, microglia activation was clearly evident in both vehicle-treated groups. Of note, anti-IBA1 staining intensities were considerably less severe at week 6 (6 weeks CPZ and Veh 36.8 ± 3.41% versus 6 weeks CPZ and LAQ 23.0 ± 1.96%), and tended to be lower at week 8 (8 weeks CPZ and Veh 23.8 ± 2.35% versus 8 weeks CPZ and LAQ 18.0 ± 2.09%) in the laquinimod-treated groups compared to the vehicle-treated groups. To determine the extent of acute axonal injury, we quantified the accumulation of synaptic vesicles by anti-APP stains (Figure 3B,D). As expected, APP^+^ spheroids were virtually absent in the control animals, whereas numerous could be found in the vehicle-treated mice. Here, the numbers were, by trend, lower in laquinimod-treated mice at week 8 (APP: CPZ and Veh 25.8 ± 5.79 spheroids/mm^2^ versus CPZ and LAQ 10.3 ± 1.86 spheroids/mm^2^).

Finally, we were interested whether laquinimod modulates the proliferation of OPC. To this end, the proportion of proliferating (Ki67^+^) oligodendrocytes (OLIG2^+^) was quantified in the different treatment groups. Low numbers of proliferating OLIG2^+^-cells were found in the control animals (~0.55%), whereas numerous were found after 6 weeks of cuprizone intoxication. However, no difference was observed between the vehicle- (~5.96%) versus laquinimod-treated (~5.91%) groups (see Appendix A).

## 4. Discussion

In this study, we have applied the following characteristic of the cuprizone model: During continuous cuprizone intoxication, remyelination fails because differentiating oligodendrocytes become vulnerable to the cuprizone toxin during differentiation. The results of in vitro studies suggest that cuprizone is selectively toxic for mature oligodendrocytes, whereas OPCs are not affected [36]. Furthermore, microglia, astrocytes, and SH-SY5Y cells were resistant to cuprizone [36,37]. We assume that once OPCs reach a certain differentiation stage, they become vulnerable against cuprizone and die, which results in the failure of remyelination during a continuous intoxication period [38]. However, if cuprizone is removed from the diet, OPCs can fully differentiate and remyelinate the affected white and grey matter brain areas. In consistence with this idea, it has been demonstrated that cuprizone retards the differentiation of oligodendrocytes in vitro [39]. Cell counts have suggested that cuprizone inhibits the maturation of oligodendrocytes without diminishing the numbers of precursors. Although it remains unclear why mature oligodendrocytes are preferentially vulnerable to cuprizone, the recent results of our lab suggest that an impaired protein folding machinery, together with stress reactions within the endoplasmic reticulum, might play an important role [40]. Oligodendrocytes protrude processes, where at the end of which, sheet-like extensions are formed, namely, myelin membranes, which ensheath axons in a multilamellar fashion to provide proper saltatory nerve conduction as well as trophic and metabolic support. Myelin membranes are unique in that approximately 70% of their dry weight consists of lipids, in particular, cholesterol, and the galactolipids galactosylceramide and sulfatide. Furthermore, myelin also contains a specific repertoire of myelin proteins, among which PLP and myelin basic protein (MBP) are the most abundant ones. All elements of this myelination machinery require a careful mutual orchestration. For example, mitochondria and the endoplasmic reticulum are two major organelles implicated in the cholesterol biosynthesis machinery. Mitochondria, for example, provide acetyl-CoAs, which are needed for cholesterol biosynthesis. On the other hand, the endoplasmic reticulum plays major roles during the propagation of secretory and membrane proteins. Since both cellular compartments, namely, the mitochondria and the endoplasmic reticulum, are functionally disturbed in the cuprizone model [40,41,42], alterations in protein–lipid trafficking or misfolding in conformational changes of myelin proteins may cause mature oligodendrocyte degeneration.

To investigate, whether newly formed oligodendrocytes are protected by laquinimod, it was essential to initiate the treatment at a time point where OPC differentiation had already started in the experimental mice. As demonstrated in Figure 2, we found low numbers of proliferating OLIG2^+^ cells in the control and 1 week cuprizone-intoxicated mice, whereas numerous were found at weeks 3 and 5. Furthermore, the quantification of APC/CC1^+^ cell numbers revealed low numbers at weeks 1 and 3, but this rapidly rebounded by the fifth week of cuprizone treatment, despite continued intoxication [43]. These data suggest that the process of remyelination, i.e., OPC proliferation and maturation, is an ongoing process between weeks 3 and 5 in this model. In line with this assumption, a recent study of our group demonstrated the presence of stressed OLIG2^+^ cells after week 5 [40], implicating that the OPCs had reached a differentiation level which had rendered them prone to the toxic effects of cuprizone. Furthermore, previous studies have indicated that the first OPCs begin to accumulate in the corpus callosum 2 weeks after initiating the cuprizone challenge, and peak between 2 to 3 weeks thereafter [30,44,45]. Thus, remyelination in the corpus callosum occurs even before demyelination is complete. Interestingly, a recent study revealed that densities of APC/CC1^+^ oligodendrocytes at 6 weeks of recovery from a cuprizone intoxication protocol were greater than that of control mice, suggesting that cuprizone-induced injury leads to differentiation of more oligodendrocytes than would be expected without injury [43]. In support of this, the density of PDGFRα^+^ OPCs at 4 weeks of cuprizone-treatment was approximately 2.5 times greater than the age-matched controls, indicating that there was a massive mobilization of progenitors after demyelination to differentiate and remyelinate the corpus callosum. To conclude, the results provided in this study and reports from others using the same model system strongly suggest that, in this study, laquinimod treatment was initiated at a time point when early remyelination cascades were already active.

Laquinimod is an immunomodulatory drug with potential anti-inflammatory and neuroprotective effects, and has been tested in MS animal models as well as in clinical studies [46,47]. Clinical trials have shown that the number of active lesions in RRMS patients are reduced and that brain volume reduction is ameliorated by laquinimod [48,49,50]. Pre-clinical studies have revealed that laquinimod has the capacity to reduce demyelination in the cuprizone model and in EAE (Experimental autoimmune encephalomyelitis) [31,32,47]. Furthermore, laquinimod prevents axonal damage, synaptic loss, and modulates immune response [31,32,51,52,53]. A number of studies suggest that laquinimod might as well be protective in other CNS pathologies than MS. For example, beneficial effects have been reported in models of Huntington’s disease [54,55,56] or in a model of traumatic brain injury [57]. However, in other experimental neurodegeneration models, such as in a model of Alzheimer’s disease, laquinimod did not show protective effects [58]. Two big clinical phase three trials were conducted to study the effectiveness of laquinimod in RRMS patients, the BRAVO (Benefit-Risk Assessment of Avonex and Laquinimod) trial [59] and ALLEGRO (Assessment of Oral Laquinimod in Preventing Progression in Multiple Sclerosis) trial [50]. In the BRAVO study, once-daily oral laquinimod resulted in statistically nonsignificant reductions in annualized relapse rate and disability progression, but significant reductions in brain atrophy versus a placebo. In the ALLEGRO trial, oral laquinimod administered once daily slowed the progression of disability and reduced the rate of relapse in patients with RRMS. Unfortunately, laquinimod failed to slow brain atrophy and disease progression in PPMS patients enrolled in the ARPEGGIO phase two clinical trial. Laquinimod was also evaluated as a potential therapy for Huntington’s disease in the LEGATO-HD phase two clinical trial [60], however, it failed to meet its primary objective of improving motor function in Huntington’s disease patients after 12 months of treatment. Most of the pre-clinical MS studies with laquinimod were performed using EAE, which recapitulates the autoimmune aspect of MS [61,62,63,64]. While auto-immune driven inflammatory demyelination is an important component of the RRMS disease stage, it is believed that autoimmunity plays a minor role in PPMS. This is best demonstrated by the finding that the classical immunomodulatory drugs [65,66] or immunosuppressive interventions [67] are largely ineffective in PPMS patients. Beyond that, histopathological studies have demonstrated that there is significantly more inflammation in SPMS (as judged by the frequency of perivascular cuffing and cellularity of the parenchyma) than in PPMS [68], and imaging studies have shown that PPMS patients have lower mean brain T2 and T1 hypointensity lesion loads than SPMS patients [69]. Why exactly the observed beneficial effects of laquinimod in pre-clinical studies did not translate into a positive impact in PPMS patients is currently unknown. On the one hand, the pathogenesis of PPMS is still largely unknown. Implicated mechanisms include the chronification of inflammation behind the relatively intact blood brain barrier, diffuse meningeal inflammation, reactive oxygen and nitrogen species produced by microglia, inducing mitochondrial dysfunction, neuronal Ca^2+^ overload, and others (see [70] for a recent review on that topic). On the other hand, classical pre-clinical animal models, especially EAE, poorly reflect the complex pathogenesis of PPMS.

In this study, we have applied a pre-clinical MS model which shares several characteristics of progressive MS, among innate driven myelin and axonal injury, functional activation of oxidative stress pathways [71], or the relative preservation of the blood-brain-barrier [72]. We were able to show that laquinimod supports remyelination in the non-supportive environment used in this model. While the results of most studies suggest that remyelination is neuroprotective [73,74,75], the results of Manrique-Hoyos and colleagues suggest that axonal degeneration continues to progress at a low level even if remyelination is complete. In their studies, animals showed an initial recovery of locomotor performance after acute cuprizone-induced demyelination. However, long after remyelination was completed (approximately 6 months after the last demyelinating episode), locomotor performance again declined in remyelinated animals as compared to the age-matched controls. This functional decline was accompanied by brain atrophy and callosal axonal loss [76]. It thus might be that laquinimod indeed restores myelin integrity in PPMS patients which does, however, not result in a superior clinical outcome or a preservation of brain atrophy. Clearly, a more complete understanding of the mechanisms involved in the pathogenesis of progressive MS phenotypes and animal models that incorporate these pathogenic characteristics is urgently needed. Comparably, further studies are needed to elucidate the underlying mechanisms of laquinimod’s protective effects in this model. As demonstrated in Figure 3, laquinimod ameliorated microglia reactivity. Microglial activation is a common feature of diverse CNS diseases, and although in some instances this activation can be damaging, the protective and regenerative functions of microglia have been revealed [77]. In the context of myelination disorders, it has been demonstrated that microglia can support remyelination in the CNS after injury via the clearance of debris, secretion of growth factors and cytokines, and through modulation of the extracellular matrix [78,79,80]. Furthermore, microglia contribute to developmental myelinogenesis and to oligodendrocyte progenitor maintenance during adulthood [78,81,82]. Beyond that, astrocytes orchestrate myelin repair by modulating microglia activity [83]. Since laquinimod can regulate glial/macrophage cell function [57,63], it might well be that parts of the observed protective effects of laquinimod in this study are due to a direct interaction with astrocytes and/or microglia.

## 5. Conclusions

In this study we were able to demonstrate that laquinimod, which has previously been shown to protect against cuprizone-induced oligodendrocyte degeneration, exerts protective effects during OPC differentiation as well. By this mechanism, laquinimod allows remyelination in non-supportive environments. These results should encourage further clinical studies in SPMS patients. Of note, laquinimod could have a significant place in the therapy of more advanced stages of disease but careful clinical studies assessing properly this potential should be conducted.

## Figures and Tables

**Figure 1 cells-08-01363-f001:**
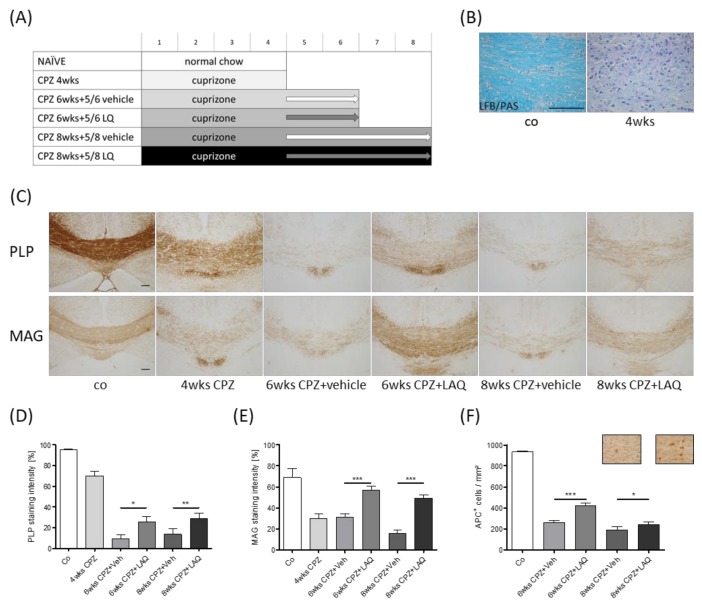
Influence of laquinimod on remyelination in the non-supportive environment. (**A**) Schematic depiction of the experimental setup. Numbers indicate the duration of the experiment in weeks. The control group is colored in white, cuprizone (CPZ) intoxication groups are colored in grayscale and black. The arrows indicate treatment with either the vehicle (Veh, light gray) or laquinimod (LAQ, dark gray) solutions. (**B**) Representative image of Luxol fast blue / periodic acid–Schiff (LFB/PAS) staining of a control animal and an animal intoxicated with cuprizone for 4 weeks. (**C**) Representative images of anti-PLP and anti-MAG stained sections of the midline corpus callosum. Densitometric analysis of (**D**) anti-PLP and (**E**) anti-MAG staining intensity (repetitive Mann–Whitney test, as indicated). (**F**) Quantification of APC^+^ cell numbers (repetitive Mann Whitney test as indicated). Representative images of anti-APC immunohistochemical stained sections of the medial corpus callosum of 8 weeks cuprizone plus vehicle (left) or laquinimod (right) treatment groups. Scale bar: 100 μm. The following symbols were used to indicate the level of significance: * *p* < 0.05, ** *p* < 0.005, and *** *p* < 0.001.

**Figure 2 cells-08-01363-f002:**
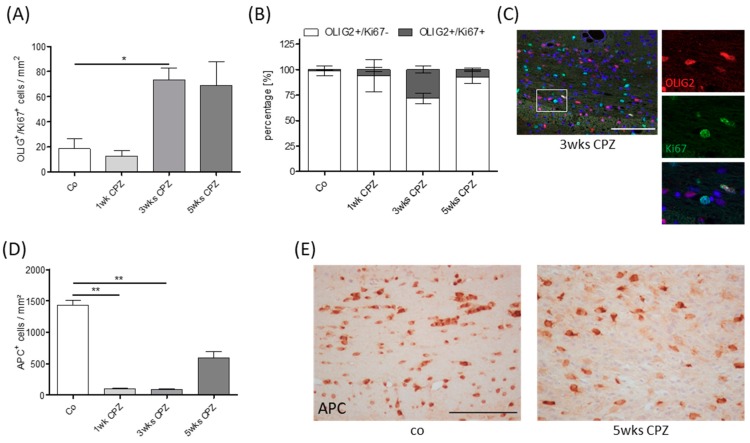
Oligodendrocyte pathology. (**A**) Quantity of OLIG2^+^ and Ki67^+^ single and double positive cells during the course of cuprizone-induced demyelination (Kruskal–Wallis test followed by Dunn’s multiple comparison test). (**B**) Percentage of single and double positive cells in relation to the entire OLIG2^+^ cell population. (**C**) Representative image of OLIG2^+^ and Ki67^+^ double positive cells. (**D**) Quantification of APC^+^ cell numbers (Kruskal-Wallis test followed by Dunn’s multiple comparison test). (**E**) Representative images of anti-APC stained sections in the medial corpus callosum. Scale bar: 100 μm. The following symbols were used to indicate the level of significance: * *p* < 0.05, ** *p* < 0.005.

**Figure 3 cells-08-01363-f003:**
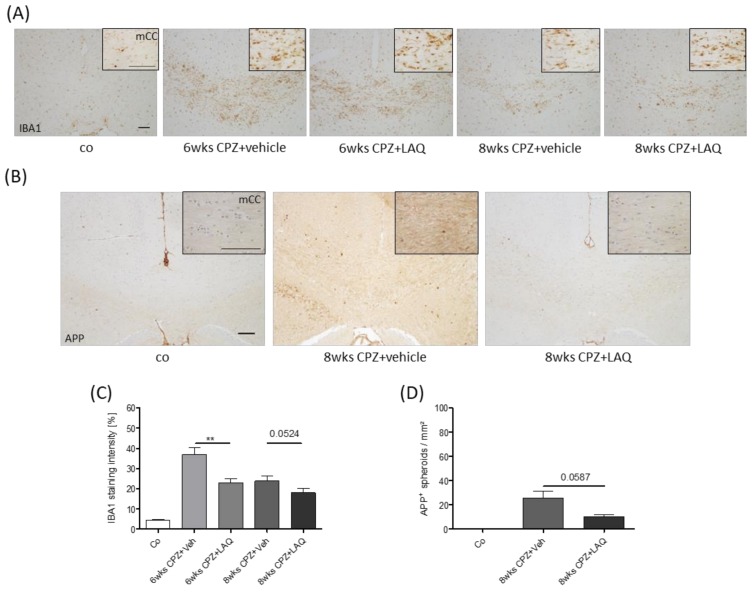
Microglia/monocyte accumulation and axonal damage. (**A**) Accumulation of microglia/monocytes visualized by anti-IBA1 immunohistochemistry. Inserts show the medial corpus callosum in higher magnification. (**B**) Axonal injury visualized by anti-APP immunohistochemistry. (**C**) Densitometric analysis of anti-IBA1 stains (Mann–Whitney test). (**D**) APP^+^ spheroid density quantification (Mann–Whitney test). Scale bar: 100μm. The following symbols were used to indicate the level of significance: ** *p* < 0.005.

**Table 1 cells-08-01363-t001:** Primary antibodies used in this study.

Name	Host/Clone	Order Number	RRID	Supplier
MAG	Mouse monoclonal	ab89780	AB_2042411	Abcam
IBA1	Rabbit polyclonal	019-19741	AB_839504	WAKO
APP (A4)	Mouse monoclonal	MAB348	AB_94882	Millipore
APC (Ab-7)	Mouse monoclonal	OP80	AB_2057371	Millipore
PLP	Mouse monoclonal	MCA839G	AB_2237198	Biorad
OLIG2	Mouse monoclonal	MABN50	AB_10807410	Millipore
Ki67	Rabbit monoclonal	ab16667	AB_302459	Abcam

**Table 2 cells-08-01363-t002:** Secondary antibodies used in this study.

Name	Host/Clone	Order Number	RRID	Supplier
Biotinylated Goat Anti-Rabbit IgG	Goat Polyclonal	BA-1000	AB_2313606	Vector Laboratories
Biotinylated Horse Anti-Mouse IgG	Horse Polyclonal	BA-2000	AB_2313581	Vector Laboratories
Donkey anti-Rabbit IgG (H+L) Highly Cross-Adsorbed Secondary Antibody, Alexa Fluor 488	Donkey Polyclonal	A21206	AB_2535792	Thermo Fisher Scientific
Goat anti-Mouse IgG2a Cross-Adsorbed Secondary Antibody, Alexa Fluor 546	Goat Polyclonal	A21133	AB_2535772	Thermo Fisher Scientific

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
