# Peer review of "Laquinimod Supports Remyelination in Non-Supportive Environments"

_cells, 2019, doi:10.3390/cells8111363_

Round 1

Reviewer 1 Report

Authors did a great job and showing a step further in discovering remyelination problems that follows MS. I have only one question/comment about the microglia involvement:

Although the cuprizone model is not suitable to study autoimmune mediated demyelination, this model is extremely helpful to elucidate basic cellular and molecular mechanisms during de- and particularly remyelination independently of interactions with peripheral immune cells. However, phagocytosis and removal of damaged myelin seems to be one of the major roles of microglia in this model and it is well known that removal of myelin debris is a prerequisite of successful remyelination. Moreover, microglia have regulatory properties potentially influencing local immune responses in the CNS and to differentially regulate MHC class II expression, promoting either effector T cell or Treg induction.

On the other hand, treatment with laquinimod has been shown to decrease Th1 and Th17 responses with a corresponding increase in CD4+CD25+FoxP3+ regulatory T cells both in the periphery and within the CNS.

It would be great to shortly mention this in the discussion part of the manuscript since authors did not discuss much the microglia response after cuprizone and laquinimod treatment.

Author Response

Thank you for the positive evaluation. We have revised the discussion section as suggested. In particular we state the following:

Comparably, further studies are needed to elucidate the underlying mechanisms of Laquinimods’ protective effects in this model. As demonstrated in figure 3, Laquinimod ameliorated microglia reactivity.  Microglial activation is a common feature of diverse CNS diseases, and although in some instances this activation can be damaging, protective and regenerative functions of microglia have been revealed [76]. In the context of myelination disorders it has been demonstrated that microglia can support remyelination in the CNS after injury via the clearance of debris, secretion of growth factors and cytokines and modulation of the extracellular matrix [77-79]. Furthermore, microglia contribute to developmental myelinogenesis and to oligodendrocyte progenitor maintenance during adulthood [77, 80, 81]. Beyond, astrocytes orchestrate myelin repair by modulating microglia activity [82]. Since Laquinimod can regulate glia/macrophage cell function [56, 62], it might well be that parts of the observed protective effects of Laquinimod in this study are due to a direct interaction with astrocytes and/or microglia. 

Reviewer 2 Report

Your work makes a solid point on the effect of Laquinimod as neuroregenerative agent utilizing your experimental model and methods. The FDA (United States) recently (March 2019) accepted the concept of “Active Secondary Progressive MS” to approve two different therapeutic molecules for this clinical variety. These medications have a selective immunosuppressive MOA with anti-inflammatory effect, but axon preservation and myelin protection do not appear to be affected. Laquinimod could have a significant place in the therapy of more advanced stages of disease but careful clinical studies assessing properly this potential, as you imply in the paper, should be implemented.

Author Response

Thank you for your positive evaluation.

We have included the following final statement at the end of the manuscript:

In this study we were able to demonstrate that Laquinimod, which has previously been shown to protect against cuprizone-induced oligodendrocyte degeneration, exerts as well protective effects during OPC differentiation. By this mechanism, Laquinimod allows remyelination in the non-supportive environment. These results should encourage further clinical studies in SPMS patients. Of note, Laquinimod could have a significant place in the therapy of more advanced stages of disease but careful clinical studies assessing properly this potential should be conducted.